# Association between tuberculosis and pregnancy outcomes: a retrospective cohort study of women in Cape Town, South Africa

Sue-Ann Meehan ,[1] Anneke C Hesseling,[1] Arne von Delft,[2,3] Florian M Marx,[1,4] Jennifer A Hughes,[1] Peter Bock,[1] Aduragbemi Banke-Thomas,[5,6] Rory Dunbar,[1] Florence Phelanyane,[2,3] Mariette Smith,[2,3] Muhammad Osman[1,5]

**Correspondence to**
Dr Sue-Ann Meehan;
sueannm@sun.ac.za

## ABSTRACT

**Background** Tuberculosis (TB) remains a leading cause of mortality among women of childbearing age and a significant contributor to maternal mortality. Pregnant women with TB are at high risk of adverse pregnancy outcomes. This study aimed to determine risk factors for an adverse pregnancy outcome among pregnant women diagnosed with TB.

**Methods** Using TB programmatic data, this retrospective cohort analysis included all women who were routinely diagnosed with TB in the public sector between October 2018 and March 2020 in two health subdistricts of Cape Town, and who were documented to be pregnant during their TB episode. Adverse pregnancy outcome was defined as either a live birth of an infant weighing <2500 g and/or with a gestation period <37 weeks or as stillbirth, miscarriage, termination of pregnancy, maternal or early neonatal death. Demographics, TB and pregnancy characteristics were described by HIV status. Logistic regression was used to determine risk factors for adverse pregnancy outcome.

**Results** Of 248 pregnant women, half (52%) were living with HIV; all were on antiretroviral therapy at the time of their TB diagnosis. Pregnancy outcomes were documented in 215 (87%) women, of whom 74 (34%) had an adverse pregnancy outcome. Being older (35–44 years vs 25–34 years (adjusted OR (aOR): 3.99; 95% CI: 1.37 to 11.57), living with HIV (aOR: 2.72; 95% CI: 0.99 to 4.63), having an unfavourable TB outcome (aOR: 2.29; 95% CI: 1.03 to 5.08) and having presented to antenatal services ≤1 month prior to delivery (aOR: 10.57; 95% CI: 4.01 to 27.89) were associated with higher odds of an adverse pregnancy outcome.

**Conclusions** Pregnancy outcomes among women with TB were poor, irrespective of HIV status. Pregnant women with TB are a complex population who need additional support prior to, during and after TB treatment to improve TB treatment and pregnancy outcomes. Pregnancy status should be considered for inclusion in TB registries.

## STRENGTHS AND LIMITATIONS OF THIS STUDY

⇒ Using the Provincial Health Data Centre (PHDC), enabled the use of routine programmatic data from multiple integrated sources, that allowed us to evaluate pregnancy and tuberculosis (TB) outcomes with consideration of HIV.
⇒ The PHDC enabled the use of an expanded definition of 'unfavourable TB treatment outcome', which more accurately reflects 'linkage to care' within the TB programme for pregnant women.
⇒ We could not determine a pregnancy outcome for 13% of our cohort, due to missing data, which is a limitation when using routine data.
⇒ The data extract used for this analysis included all persons with TB, which meant that we were not able to compare pregnancy outcomes for women who did not have TB.

## BACKGROUND

Globally, 10.6 million people developed tuberculosis (TB) in 2021, of whom 3.4 million (32.1%) were women.[1] TB incidence peaks in women during the reproductive years,[2 3] is a leading cause of mortality among women of childbearing age[4] and a significant contributor to maternal mortality.[5] Pregnant women with TB are at high risk of adverse pregnancy outcomes including prematurity, low birth weight and stillbirth.[6–8]

TB is common in settings with high HIV prevalence, especially in sub-Saharan Africa. People living with HIV have a 2.5 times increased risk of developing TB despite access to antiretroviral therapy (ART),[9] and TB remains the leading cause of death among people living with HIV.[10] Pregnant women with TB and HIV are at increased risk of adverse pregnancy outcomes[11 12] and higher maternal and infant mortality.[13–16] There are limited data on TB treatment outcomes among pregnant women since pregnancy status is not routinely captured or reported as a TB programme indicator globally and pregnancy testing results are not routinely captured for women with TB. While ART[17]

and TB preventive treatment (TPT)[18] reduces the risk of TB disease, an estimated 150 600 pregnant and 49 000 postpartum women develop TB annually, with women in Africa accounting for 40% of this burden.[19] There is a need to integrate routine pregnancy indicators in TB programmes to improve surveillance of pregnant women with TB.

In high TB burden settings, the WHO recommends screening pregnant women for TB at every contact with a healthcare worker.[20] In South Africa, routine testing of all pregnant women for TB and providing TPT once TB disease is excluded is recommended[18] but the extent of implementation is unknown and most likely incomplete. There is a pressing need for improved data from routine and programmatic sources to identify programmatic gaps and inform evidence-based research interventions to improve health outcomes for pregnant women with TB and their infants. This study used routine data across three health programmes (TB, maternal health and HIV services) to determine risk factors associated with an adverse pregnancy outcome for women with TB.

## METHODS
### Setting
We used routine health data from two large health subdistricts (Khayelitsha and Tygerberg) in the Cape Metro, Western Cape Province, South Africa. The Cape Metro has a population of approximately 4.6 million people (two-thirds of the provincial population); HIV/AIDS and TB were among the top six causes of death in 2015.[21] Antenatal HIV prevalence was 22% in 2019.[22] There were 26 000 newly diagnosed persons with TB and more than 2000 TB deaths in 2022.[23] Maternal mortality ranged between 43.6 and 66.8 per 100 000 live births at healthcare facilities in 2017–2019.[24] Khayelitsha is a peri-urban low resourced health subdistrict, with a mix of formal and informal housing.[25] It is serviced by a district hospital and 10 primary healthcare (PHC) facilities.[21] Tygerberg is a large health subdistrict,[25] with mostly formal dwellings. It has a tertiary hospital, a district level hospital and 21 PHC facilities.[21]

### Study design
This was a retrospective cohort study based on programmatic data collected as part of a larger health system strengthening study (LINKEDin), which aimed to reduce initial loss to follow-up for people diagnosed with TB in three South African provinces.[26]

### Study population
We included data for all women who were routinely diagnosed with TB at either a hospital or PHC facility in the two health subdistricts between October 2018 and March 2020 and who were documented as having a pregnancy that overlapped with their TB episode (see table 1 for definition of key terms), as recorded in routine health records.

| Table 1 | Key variable definitions for TB, HIV and pregnancy outcomes |
|---|---|
| **Definition of terminology** | |
| TB episode | Period from the date of TB diagnosis to the recorded date of TB outcome or death. |
| TB diagnosis | Includes a bacteriological (evidence of a positive TB laboratory test) or clinical diagnosis of TB (including evidence of TB drugs administered as part of a therapeutic regimen, or primary ICD-10 codes and inclusion in a primary care TB register). |
| Pregnancy episode | The period from the first electronically captured evidence of a pregnancy documented in antenatal health services to the date of delivery or of any other pregnancy outcome. |
| Overlapping pregnancy | Date of the first evidence of the pregnancy was ≤9 months prior to the date of the TB diagnosis or ≤18 months after the date of the TB diagnosis. |
| **TB treatment outcomes** | |
| Favourable TB outcome | Recorded in the routine health service data as treatment success (bacteriologically cured or successfully completed TB treatment). |
| Unfavourable TB outcome (expanded definition) | Recorded in the routine health service data as lost to follow-up, died during treatment, treatment failed or not evaluated (WHO TB outcomes) OR loss to follow-up prior to linkage to TB care (including died prior to linkage, that is, 'initial loss to follow-up'). |
| Linkage to TB care | Evidence of accessing a TB treatment facility for TB treatment (TB hospital or a PHC facility offering TB treatment) after the diagnosis of TB. |
| **Pregnancy outcomes** | |
| Good pregnancy outcome | A pregnancy that resulted in the live birth of an infant weighing ≥2500 g and with a gestation period ≥37 weeks. |
| Adverse pregnancy outcome | A pregnancy that resulted in either a live birth of an infant weighing <2500 g and/or with a gestation period <37 weeks or where there was a stillbirth, miscarriage, termination, the mother dying prior to delivery or early neonatal death. |

ICD-10, Diagnostic coding standard; PHC, primary healthcare; TB, tuberculosis.

## Data collection

Data were extracted from the Western Cape Provincial Health Data Centre (PHDC). The PHDC is a comprehensive linked health information exchange in the province and uses unique identifiers to integrate all electronic health data from routine health information systems in the public sector (laboratory, pharmacy, administrative and other clinical data) into single patient level records to support improved patient care.[27] PHDC outputs include line lists (cascades) of patients with specific conditions, for example, TB, HIV, pregnancy.[27] These are updated daily and can be used to track patients at each healthcare visit and identify those requiring follow-up, including those newly diagnosed with TB who have not yet been entered into a TB treatment register and initiated on TB treatment.

We accessed the de-identified TB cascade from the PHDC and filtered for pregnancy status, to identify all women with TB who were pregnant during the study period in the two subdistricts. We used the unique identifier for these individuals to access data contained in the HIV and pregnancy cascades. Cascades are individual patient level views of specific health conditions which include key dates, evidence, outcomes and comorbidities relevant for patient management. We merged the three cascades and checked our data set for consistency, confirming that all women included in our cohort had a pregnancy that overlapped with their TB episode (table 1).

We extracted demographic (age) and clinical variables for TB, HIV and pregnancy (eg, *Mycobacterium tuberculosis* drug susceptibility status, HIV status, CD4 count, gravidity, infant birth weight, etc) and key dates (date of TB diagnosis, date first known to the public antenatal services, date of delivery, etc). To determine TB treatment outcomes, we reviewed the evidence recorded in the PHDC relating to evidence of linkage to a TB treatment facility and if linked, we extracted their TB treatment initiation and outcome data (cured, successfully completed, treatment failure, loss to follow-up, died, not evaluated). We allocated a TB treatment outcome (favourable or unfavourable) as per our definition (table 1). We used an expanded definition of unfavourable TB outcome to include newly diagnosed persons with TB who did not link to care and initiate treatment at a TB treatment facility. To determine a pregnancy outcome, we reviewed live birth, infant birth weight and gestational period and allocated a pregnancy outcome (good or adverse), as per definitions in table 1. Women who had one or more of these variables missing, had a pregnancy outcome of unknown assigned. Pregnancy outcomes could occur either during or after the end of the TB episode (figure 1).

## Power calculation

As we were limited to a fixed sample size for this study, we implemented a pragmatic approach and considered the high proportion of adverse pregnancy outcomes in women with TB, ranging from 65% in a small cohort at

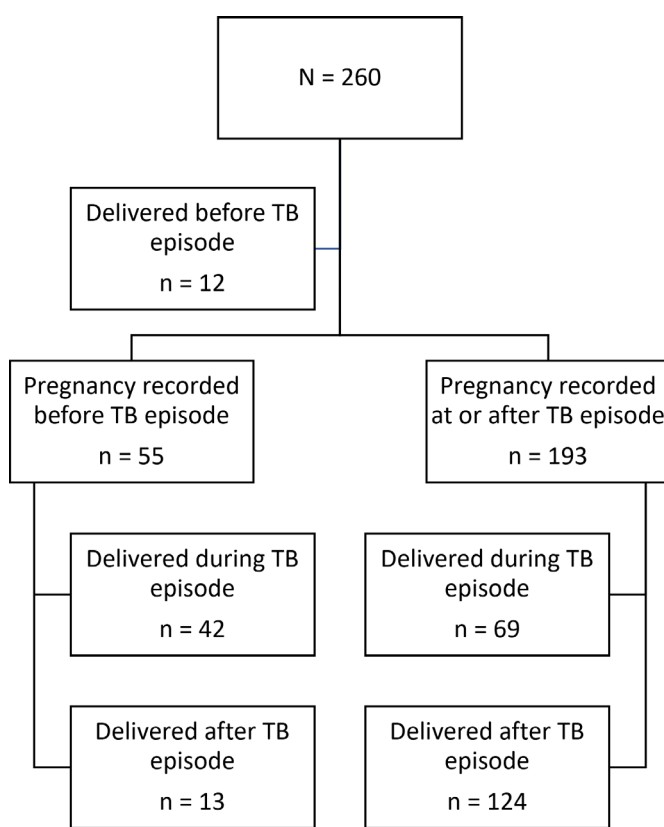

**Figure 1** Timing of pregnancy in relation to the tuberculosis episode. TB, tuberculosis.

a specialised hospital[15] to 27% in a systematic review and meta-analysis of pregnant women with drug-resistant TB.[6] With no data on pregnancy outcomes by TB treatment outcomes, we assumed a difference and estimated our sample of 248 pregnant women with TB would have 89% power with 95% significance to detect a difference of 20% points in adverse pregnancy outcomes by TB outcomes.

## Patient and public involvement

As this study used routine data at the programme level, patients were not directly involved in any aspect of the study. We did engage our institutional community advisory board (CAB), on the design and implementation strategies of the broader overall study (LINKEDin), which primarily involved the evaluation of two interventions to reduce initial loss to follow-up among patients with TB in South Africa. In this subanalysis we used standardised TB and pregnancy outcomes. We presented our findings to the CAB prior to writing this manuscript and then again more recently to get their input on a proposed future study to address poor TB and pregnancy outcomes among this population.

## Statistical analysis

Descriptive statistics were used to describe baseline characteristics, the timing of pregnancy and delivery compared with the TB episode, stratified by HIV status. A logistic regression model was used to determine risk factors for adverse pregnancy outcomes including TB treatment

outcomes and other predictors. Predictors were added incrementally, observing the change in significance of the likelihood ratio test of each model, to produce a final adjusted model. The relationship between predictors was considered and we avoided co-linearity in the final model. SAS software (V.9.4; SAS Institute, Cary, North Carolina, USA) was used for analysis.

### Ethical considerations

The Health Research Ethics Committee of Stellenbosch University (N18/07/069) approved the study, which was conducted according to the guiding principles within the Declaration of Helsinki. Approvals were received from the Western Cape Department of Health and the City of Cape Town Health Directorate. We received a waiver of individual consent for this routinely collected health data.

## RESULTS

### Overall

A total of 260 pregnant women were diagnosed with TB between October 2018 and March 2020, of whom 67 had a pregnancy recorded prior to their TB diagnosis. We excluded 12 women because their pregnancy outcome was also recorded prior to their TB diagnosis. There were 111 women who had their pregnancy outcome documented before their TB outcome (ie, who delivered while still on TB treatment), and 137 women who had their pregnancy outcome documented after their TB outcome (ie, who delivered after their TB treatment). See figure 1.

Among the 248 women included in the analysis, the median age was 27.7 years (IQR: 22.7–32.3), range 14–43 years. The majority of women (191/248; 77%) had bacteriologically confirmed TB, 170/248 (69%) were treated for drug-susceptible TB and 10/248 (4%) were treated for drug-resistant TB. All had an HIV status recorded, half (130/248; 52%) were living with HIV. Women living with HIV (WLWH) were slightly older (29.1 years; IQR: 25.7–33.4) compared with HIV negative women (25.4 years; IQR: 20.6–30.1). Of WLWH, the median CD4 count at TB diagnosis was 236 cells/µL (IQR: 127–423) and all were on ART at the time of their TB diagnosis, with 56/130 (43%) having initiated ART during their pregnancy. Online supplemental table S1 provides characteristics of pregnant women with HIV by pregnancy status at the time ART was initiated. Overall, 9/248 of women (4%) died, of whom 6/9 (66%) were WLWH. Most women (6/9) died >60 days after their TB diagnosis had been made (table 2).

For half of the women (121/248; 49%), this was their first pregnancy. Most (193/248; 78%) had their pregnancy recorded at or after the time their TB diagnosis was made, with 64% (124/193) delivering after their TB outcome was documented (figure 1). One-third (83/248; 33%) were documented to have attended antenatal services late, presenting for the first visit within 30 days of their delivery, with most of these women (67/83; 81%) presenting to services within 2 days of delivering their

infant. Overall, the median gestation time was 40 weeks (IQR: 35–40 weeks). Among infants overall, the median birth weight was 2875 g (IQR: 2445–3270 g) (table 2).

### TB treatment outcomes

Using the more comprehensive definition including initial loss to follow-up, one-third (79/248; 32%) of pregnant women had an unfavourable TB treatment outcome; 44/130 (33.8%) among WLWH and 35/118 (29.7%) among HIV negative women. Of those with an unfavourable TB treatment outcome, 34% (27/79) were never linked to TB care and were not recorded to have started TB treatment in a TB treatment register. Of those not linked to care, a quarter (7/27; 26%) died (3 HIV negative and 4 HIV positive) (figure 2).

### Pregnancy outcomes

Pregnancy outcomes were documented in 215/248 (86.7%) women. Overall, 74/215 (34.4%) women had an adverse pregnancy outcome; 40/106 (37.7%) among WLWH and 34/109 (31.2%) among HIV negative women. Overall, adverse pregnancy outcomes were higher among women who had unfavourable TB treatment outcomes (32/79; 40.5%) versus favourable TB treatment outcomes (42/169; 24.9%), irrespective of HIV status (figure 2).

### Risk factors associated with an adverse pregnancy outcome

Pregnant women with TB were at increased odds of having an adverse pregnancy outcome if they were older (adjusted OR (aOR) 3.99: 95% CI: 1.37 to 11.57), living with HIV (aOR 2.72: 95% CI: 1.2 to 6.18), had an unfavourable TB treatment outcome (aOR 2.29: 95% CI: 1.03 to 5.08) and if they presented late to antenatal services (≤30 days prior to their delivery), compared with attending antenatal services for at least 121 days (aOR 10.57: 95% CI: 4.01 to 27.89). Women were less likely to have an adverse pregnancy outcome with increasing gravidity (aOR 0.55: 95% CI: 0.37 to 0.81) (table 3).

## DISCUSSION

This study included a large cohort of women diagnosed with TB in routine services with a pregnancy that overlapped their TB episode. Pregnancy outcomes were poor; being older, living with HIV, having an unfavourable TB treatment outcome and presenting late to antenatal services were all risk factors associated with an adverse pregnancy outcome. Half of these women were living with HIV, higher than the estimated antenatal HIV prevalence rate of 22% in the setting.[22] All HIV-positive women were on ART; more than half (57%) were already on ART prior to their pregnancy, with the balance initiating ART during their pregnancy. Overall, these findings indicate good HIV screening and ART initiation services generally and specifically among pregnant women attending public antenatal services. As comparison, approximately 54% of all people living with HIV were on ART in 2019 in Cape Town.[28]

**Table 2** Demographic and clinical characteristics among pregnant women with tuberculosis, by HIV status (n=248)

| | Total<br>N (%) | HIV status<br>n (%) | |
|---|---|---|---|
| | (N=248) | HIV+ (n=130) | HIV− (n=118) |
| Age in years, median (IQR) | 27.7 (22.7–32.3) | 29.1 (25.7–33.4) | 25.4 (20.6–30.1) |
| 14–17 years | 10 (4.1) | 3 (2.3) | 7 (5.7) |
| 18–24 years | 75 (30.2) | 25 (19.2) | 50 (40.7) |
| 25–34 years | 129 (52.0) | 82 (63.1) | 47 (38.2) |
| 35–44 years | 34 (13.7) | 20 (15.4) | 14 (11.4) |
| Pregnancy recorded in relation to TB diagnosis | | | |
| Pregnant before TB diagnosed | 55 (22.2) | 33 (25.4) | 22 (18.6) |
| Pregnant after TB diagnosed | 193 (77.8) | 97 (74.6) | 96 (81.4) |
| TB diagnosis (%) | | | |
| Bacteriologically confirmed | 191 (77.0) | 93 (71.5) | 98 (83.1) |
| Clinically diagnosed | 57 (23.0) | 37 (28.5) | 20 (16.9) |
| Place of TB diagnosis (%) | | | |
| Hospital | 78 (31.5) | 46 (35.4) | 32 (27.1) |
| PHC facility | 170 (68.5) | 84 (64.6) | 86 (72.9) |
| TB treatment category | | | |
| New | 162 (65.3) | 85 (65.4) | 77 (65.3) |
| Retreatment | 66 (26.6) | 37 (28.5) | 29 (24.6) |
| Unknown | 20 (8.1) | 8 (6.2) | 12 (10.2) |
| Site of disease | | | |
| Extrapulmonary TB | 44 (17.7) | 32 (24.6) | 12 (10.2) |
| Pulmonary TB | 196 (79.0) | 92 (70.8) | 104 (88.1) |
| Not specified | 8 (3.2) | 6 (4.6) | 2 (1.7) |
| Drug susceptibility status | | | |
| DR TB | 10 (4.0) | 5 (3.8) | 5 (4.2) |
| DS TB | 170 (68.5) | 81 (62.3) | 89 (75.4) |
| DST not recorded | 68 (27.4) | 44 (33.8) | 24 (20.3) |
| TB outcomes (%) | | | |
| Favourable | 169 (68.1) | 86 (66.2) | 83 (70.3) |
| Unfavourable | 79 (31.9) | 44 (33.8) | 35 (29.7) |
| Linked to TB care | | | |
| Linkage ≤30 days | 202 (81.5) | 107 (82.3) | 95 (80.5) |
| Linkage >30 days | 19 (7.7) | 11 (8.5) | 8 (6.8) |
| Unlinked | 27 (10.9) | 12 (9.2) | 15 (12.7) |
| Died | | | |
| Yes | 9 (3.6) | 6 (4.6) | 3 (2.5) |
| No | 239 (96.4) | 124 (95.4) | 115 (97.5) |
| Time to maternal death (from date of TB diagnosis) | | | |
| 0–30 days | 3 (33.3) | 1 (16.7) | 2 (66.7) |
| 30–59 days | 0 (0) | 0 (0) | 0 (0) |
| 61–90 days | 2 (22.2) | 2 (33.3) | 0 (0) |
| ≥90 days | 4 (44.4) | 3 (50.0) | 1 (33.3) |
| Parity | | | |
| 0 | 121 (48.8) | 55 (42.3) | 66 (55.9) |

Continued

**Table 2** Continued

| | Total N (%) | HIV status n (%) | |
|---|---|---|---|
| | (N=248) | HIV+ (n=130) | HIV− (n=118) |
| 1 | 71 (28.6) | 47 (36.2) | 24 (20.3) |
| 2 | 35 (14.1) | 19 (14.63.8) | 16 (13.6) |
| >2 | 21 (8.5) | 9 (6.9) | 12 (10.2) |
| Gravidity | | | |
| 1 | 116 (46.8) | 52 (40.0) | 64 (54.2) |
| 2 | 71 (28.6) | 46 (35.4) | 25 (21.2) |
| 3 | 39 (15.7) | 21 (16.2) | 18 (15.3) |
| >3 | 22 (8.9) | 11 (8.5) | 11 (9.3) |
| Gestation (weeks), median (IQR) | 40 (35–40) | 40 (35–40) | 40 (35–40) |
| Gestation (weeks) | | | |
| <30 | 14 (5.6) | 8 (6.2) | 6 (5.1) |
| 30–40 | 228 (91.9) | 119 (91.5) | 109 (92.4) |
| >40 | 5 (2.0) | 3 (2.3) | 2 (1.7) |
| Not recorded | 1 (0.4) | 0 (0) | 1 (0.8) |
| Birth weight, in grams, median (IQR) | 2870 (2445–3270) | 2835 (2400–3320) | 2902 (2500–3240) |
| Birth weight (g) | | | |
| Very low/low (500–2499) | 56 (22.6) | 31 (23.8) | 25 (21.2) |
| Normal (2500–3900) | 142 (57.3) | 66 (50.8) | 76 (64.4) |
| >3900 | 4 (1.6) | 2 (1.5) | 2 (1.7) |
| Unknown | 46 (18.5) | 31 (23.8) | 15 (12.7) |
| Time known to ANC services (days) (%) | | | |
| 0–30 | 83 (33.5) | 33 (25.4) | 50 (42.4) |
| 31–120 | 70 (28.2) | 41 (31.5) | 29 (24.6) |
| ≥121 | 95 (38.3) | 56 (43.1) | 39 (33.1) |
| Pregnancy outcome (%) | | | |
| Good | 141 (56.9) | 66 (50.8) | 75 (63.6) |
| Adverse | 74 (29.8) | 40 (30.8) | 34 (28.8) |
| Unknown* | 33 (13.3) | 24 (18.5) | 9 (7.6) |

*Excluded from the regression model used to predict adverse pregnancy outcomes, as there was missing data for one or more of the following variables; live birth status, birth weight or gestational age.
ANC, antenatal care; DR, drug-resistant; DS, drug-susceptible; DST, drug-susceptible test; PHC, primary healthcare; TB, tuberculosis.

Overall, 34% of women with TB had an adverse pregnancy outcome. A systematic review of 69 studies (published 2009–2021), the majority of which were from low-income and middle-income countries, reported poor pregnancy outcomes for women with TB, with most studies reporting low birth weight and/or preterm labour and/or spontaneous abortions.[29] A South African study reported 48% of women routinely treated for multi-drug resistant/rifampicin resistant TB (MDR/RR-TB) during 2013–2017, had a poor pregnancy outcome,[30] higher than our study, where the majority had drug-sensitive TB. Other South African studies with cohorts between 2011[15] and 2014[14] showed that of infants born to HIV-positive mothers with TB, between 20% and 70% had a gestation period <37 weeks and between 21% and 59% had a low birth weight. Since these previous studies, there have been many substantial advances and positive policy changes in our setting including universal HIV testing, universal ART rollout[31] and pregnant women being eligible for TB preventive therapy irrespective of HIV status.[18] Yet our findings show similar high rates of poor pregnancy outcomes in women, irrespective of HIV status. This is highly concerning. Our data highlights the need for additional support and interventions for pregnant women with TB, including strategies to improve the integration of TB and maternal services to improve TB and pregnancy outcomes for these women. Additional work is needed to understand the care pathway for these

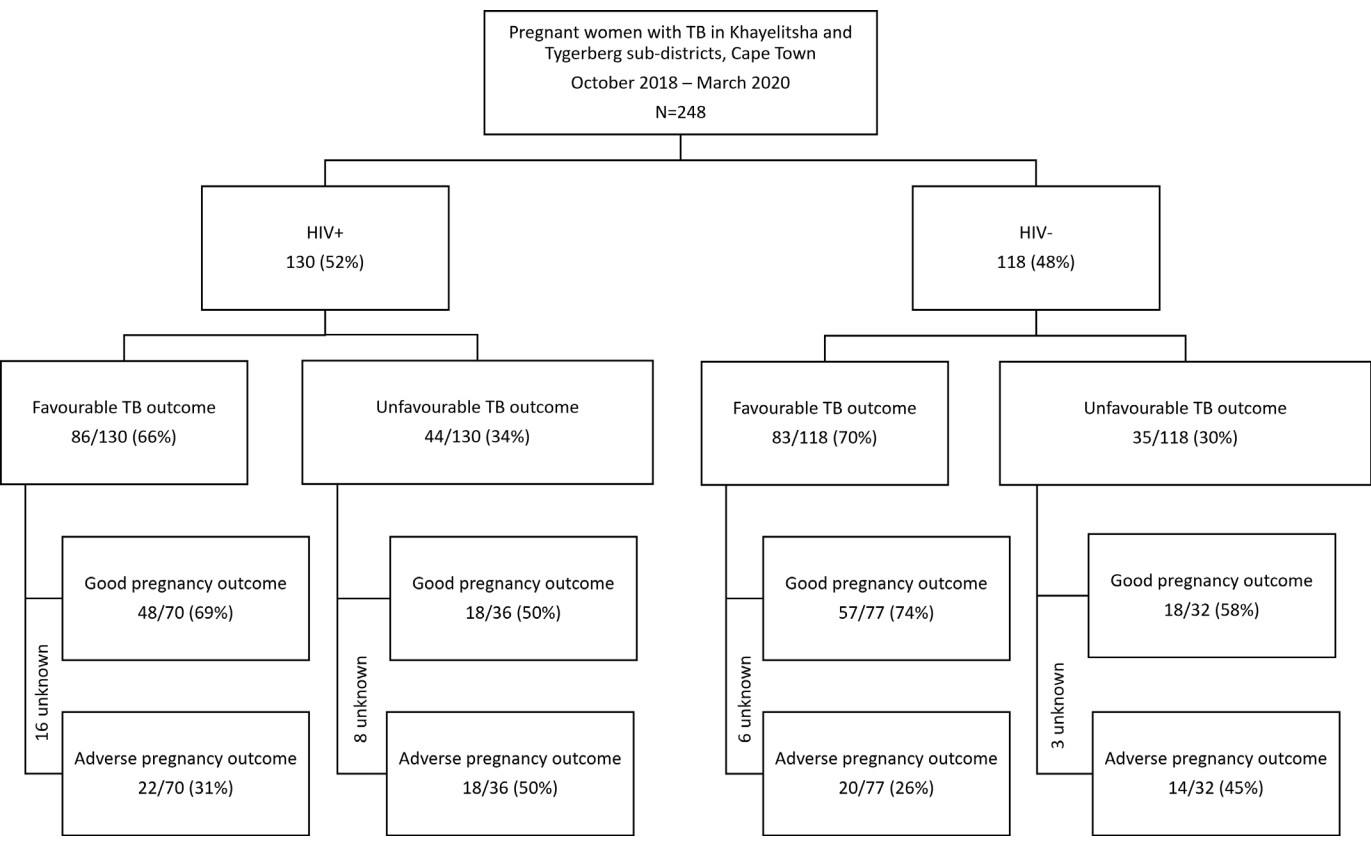

**Figure 2** Tuberculosis outcomes and pregnancy outcomes by HIV status (n=248). TB, tuberculosis.

women, barriers and enablers to care and determine gaps between policy and implementation. We also note that comparison across studies is challenging as the definition of an adverse pregnancy outcome varied by study. A standardised definition including all variables that affect both maternal and infant outcomes is needed.

Having a poor TB treatment outcome was associated with having a poor pregnancy outcome, which is biologically plausible. We are not aware of other studies that have reported on poor TB treatment outcomes as a risk factor for adverse pregnancy outcomes. A study among pregnant women with MDR/RR-TB found that 40% had an unfavourable TB treatment outcome, as per traditional WHO definitions.[7] Our study found one-third of women had an unfavourable TB outcome, but used a broader definition to include those lost prior to registration/treatment initiation.[32] We identified a high proportion of women who did not link to TB care; 43% among HIV-negative women and 27% among WLWH. For persons with TB in South Africa, approximately 12% among drug-sensitive[33] and 37% among drug-resistant[34] persons with TB are never started on treatment. Considering that the majority of our cohort had drug-sensitive TB, loss prior to registration/treatment initiation among our cohort was higher than what is reported for persons with TB overall. As these women are not recorded in TB treatment registers, their data is not included in routine recording and reporting, further exacerbating the gaps in surveillance of TB in women of childbearing age.

Pregnant women diagnosed with TB are a complex population and require more intensive follow-up to ensure better TB and pregnancy outcomes, and to reduce mortality. We recommend improved one-on-one communication by healthcare workers when referring these women to care.[35] Skilled personnel should ensure that women understand their disease and the importance of linking to ongoing care and treatment. More broadly, health services should update contact details of persons with TB at each health visit, so that they can be easily traced if they do not link to TB care. This is especially important for pregnant women with TB, who may have additional barriers to accessing care. Routine antenatal and postnatal services should serve as an important trigger to link women to TB treatment following screening and testing. Being lost to the health system after a TB diagnosis may be the underlining reason for having an adverse pregnancy outcome. The association between TB and pregnancy outcomes should be further explored to better understand how these are associated.

Most women had their pregnancy recorded after they were diagnosed with TB. This is an interesting finding, as typically women are screened for TB during antenatal visits. This finding may be highlighting a need for improved sexual and reproductive health counselling with women who are accessing TB care in this setting. We are also mindful of the limitations of using routine data and that the capturing of the first date known to the antenatal services may not always be accurate. This

Table 3 Multivariable logistic regression model for predictors of adverse pregnancy outcomes among women with tuberculosis (n=215)

| | Total | Adverse | % | OR (95% CI) | aOR (95% CI) |
|---|---|---|---|---|---|
| Age category | | | | | |
| 14–17 years | 8 | 2 | 25.0 | 0.76 (0.15 to 3.96) | 0.5 (0.05 to 5.24) |
| 18–24 years | 64 | 21 | 32.8 | 1.12 (0.58 to 2.15) | 1.56 (0.65 to 3.76) |
| 25–34 years | 115 | 35 | 30.4 | Reference | |
| 35–44 years | 28 | 16 | 57.1 | 3.05 (1.31 to 7.11) | 3.99 (1.37 to 11.57) |
| HIV status | | | | | |
| HIV+ | 106 | 40 | 37.7 | 1.34 (0.76 to 2.35) | 2.72 (1.2 to 6.18) |
| HIV– | 109 | 34 | 31.2 | Reference | |
| Location of diagnosis | | | | | |
| Hospital | 64 | 32 | 50.0 | 2.6 (1.42 to 4.76) | 2.14 (0.99 to 4.63) |
| PHC | 151 | 42 | 27.8 | Reference | |
| Previous TB treatment history | | | | | |
| Retreatment | 55 | 20 | 36.4 | 1.19 (0.62 to 2.29) | 0.7 (0.31 to 1.57) |
| New | 142 | 46 | 32.4 | Reference | |
| Diagnostic method (a) | | | | | |
| Bacteriologically confirmed | 168 | 64 | 38.1 | 2.28 (1.06 to 4.89) | 2.33 (0.86 to 6.29) |
| Clinically diagnosed | 47 | 10 | 21.3 | Reference | |
| Site of disease (a) | | | | | |
| PTB | 174 | 59 | 33.9 | Reference | |
| EPTB | 37 | 14 | 37.8 | 1.19 (0.57 to 2.47) | |
| DST (a) | | | | | |
| DR TB | 8 | 4 | 50.0 | 1.7 (0.41 to 7.05) | |
| DST not recorded | 56 | 14 | 25.0 | 0.57 (0.28 to 1.13) | |
| DS TB | 151 | 56 | 37.1 | Reference | |
| TB treatment linkage (b) | | | | | |
| Linked within 30 days | 172 | 52 | 30.2 | Reference | |
| Linked after 30 days | 19 | 6 | 31.6 | 1.07 (0.38 to 2.96) | |
| Never linked | 24 | 16 | 66.7 | 4.62 (1.86 to 11.45) | |
| TB treatment outcome (b) | | | | | |
| Unfavourable | 68 | 32 | 47.1 | 2.22 (1.23 to 4.03) | 2.29 (1.03 to 5.08) |
| Favourable | 147 | 42 | 28.6 | Reference | |
| Time attending antenatal services | | | | | |
| 0–30 days | 71 | 36 | 50.7 | 4.87 (2.36 to 10.06) | 10.57 (4.01 to 27.89) |
| 31–120 days | 58 | 23 | 39.7 | 3.11 (1.45 to 6.69) | 3.54 (1.39 to 8.98) |
| ≥121 days | 86 | 15 | 17.4 | Reference | |
| Parity (c) | | | | | |
| Continuous | | | | 0.7 (0.52 to 0.95) | |
| Gravidity (c) | | | | | |
| Continuous | | | | 0.7 (0.53 to 0.94) | 0.55 (0.37 to 0.81) |

(a), (b) and (c) not added simultaneously to final model to avoid collinearity.
aOR, adjusted OR; DR, drug-resistant; DS, drug-susceptible ; DST, drug-susceptible test; EPTB, extrapulmonary tuberculosis; PHC, primary healthcare; PTB, pulmonary tuberculosis; TB, tuberculosis.

finding should be interpreted with caution. Offering effective contraception to women at TB treatment initiation[29 36] and screening for pregnancy at each TB-related visit, with integration of reproductive health services as part of TB care is critical.[37] Women should be made aware of their increased risk of poorer pregnancy outcomes and need for additional care if they conceive during their TB episode. If adequate counselling and assessment of family planning at the start of TB treatment is being implemented as per local guidelines, and women continue to fall pregnant during their TB episode, then there could be other factors associated with this. Socio-behavioural research may elicit such factors and provide more person-centred understanding that could guide better implementation of existing or more patient-centred policies.

Living with HIV was found to be a risk factor for having an adverse pregnancy outcome, similar to previous studies. In our study, women had similar poor pregnancy outcomes irrespective of whether ART was started prior to or during pregnancy. A systematic review of 73 cohort studies, found that ART reduced the risk of adverse perinatal outcomes in pregnant women living with HIV, but the risk remained higher than in HIV-negative women.[38] This review did not specify if cohorts had TB during their pregnancy. There exists limited data on HIV as a risk factor for adverse pregnancy outcomes among women with TB. A South African study from a tertiary hospital comparing TB-exposed infant outcomes by HIV-exposure, found a higher proportion of prematurity (70% vs 52%), low birth weight (59% vs 57%) and infant mortality (4 stillbirths and 6 neonatal deaths vs 0 deaths) among HIV-exposed infants compared with HIV-unexposed infants.[15] In this study 64% of women were on ART, compared with our study where all women were on ART. ART may reduce the risk of adverse pregnancy outcomes, but for WLWH and those who have TB overlapping their pregnancy, adverse pregnancy outcomes are high, irrespective of ART. Pregnancy remains an excellent opportunity to start ART, the WHO recommends the immediate initiation of lifelong ART for all people living with HIV.[39] Despite a universal test and treat policy for HIV, the findings from our study show that many women start ART only at antenatal presentation. An increase in community-based HIV testing and treatment to support earlier ART initiation is essential. ART initiation prior to pregnancy may reduce adverse pregnancy outcomes for these women and also TB treatment outcomes.

Our findings show that one-third of women (25% of WLWH and 42% of HIV-negative women) were only documented to have sought care at public antenatal services during the month prior to delivery, highlighting the challenge of 'late booking'. All antenatal, TB and HIV services are free of direct charge in South Africa in the public sector. This proportion is higher than in the 2019 South African Antenatal survey, where only a small proportion of women (5.6%) attended their first antenatal visit in the last trimester of their pregnancy.[40] In the Cape Town Metro, 75% of women attend antenatal services before 20 weeks.[22] South Africa guidelines indicate that booking should be as early as possible[41] to ensure optimal antenatal care. Clearly, pregnant women with TB are an especially high-risk group overall. Those who book late should be tested for TB as a priority and offered TPT. Tracking pregnancy using 'action lists' may offer the opportunity for personnel in the TB programme to create awareness around the importance of adequate antenatal visits and urge these women to access maternal services earlier in their pregnancy.

We found that gravidity had a protective effect. This could be an example of the 'healthy mother effect', where women who have had successful pregnancies in the past are more likely to have successful pregnancies in the future.[42] This finding should be interpreted with caution.

Our study addresses a key yet neglected aspect of TB and maternal child health, using routine health data from multiple integrated sources that allowed us to evaluate pregnancy and TB treatment outcomes with consideration of HIV. We demonstrate poor pregnancy outcomes in pregnant women with TB, half of whom were also living with HIV. Using the PHDC strengthened our ability to determine more inclusive definitions of favourable and unfavourable TB treatment outcomes and enabled us to include all pregnant women diagnosed with TB, so we could determine both pre-loss and post-loss to follow-up during TB care. Using our expanded definition of an 'unfavourable' TB treatment outcome, this study was able to reflect more accurately 'linkage to care' within the TB programme for pregnant women. From a programme perspective, having a maternal cascade is a benefit for the monitoring of the maternal and infant health programme in services. Health services should allow for integrated tools to provide optimal care for patients across multiple programmes: maternal, TB and HIV programmes.

We could not determine a pregnancy outcome for 13% of our cohort, due to missing data. This may be due to women delivering outside the province, data linkage limitations between mother and infant or incomplete data capture, potentially resulting in a biased outcome and over-reporting of adverse pregnancy outcomes. Missing data is a limitation when using routine health data sources. A second limitation is that we were unable to compare pregnancy outcomes for our cohort to women who did not have TB during their pregnancy. This analysis was nested within a larger study and our data extract was restricted to persons with TB. Future work should compare pregnancy outcomes for women by TB status and TB outcomes by pregnancy status. A third limitation is that while the multivariable regression model was able to control for selected factors, residual confounding could have occurred due to other maternal factors (e.g. gestational diabetes, hypertension, anaemia) or socioeconomic factors (e.g. poverty, substance use, educational status). These data were not available for analysis and should be included in future work. A further limitation is the inability to explain the time between presenting with TB and the timing of pregnancy.

## CONCLUSIONS

Pregnancy outcomes among women with TB were poor, irrespective of HIV status. This is a major concern. TB care needs to be strengthened for pregnant women and health services need to think through how best to optimise maternal services within the TB programme. We used routine data to track women across the TB, HIV and maternal healthcare cascades, allowing for more inclusive definitions and analysis of linkage to care and TB treatment and pregnancy outcomes. Our study highlights the high proportion of women diagnosed with TB who never started TB treatment and the large proportion of women who only sought antenatal care services during the month prior to delivery. Pregnant women with TB are a complex population who need additional support prior to, during and after TB treatment to improve pregnancy outcomes, irrespective of HIV status. Going forward, recording of pregnancy status should be a key indicator piloted as part of TB treatment registries, as this would result in better quality data and offer opportunities for earlier intervention and support of pregnant and postpartum women with TB.

**Author affiliations**
$^1$Desmond Tutu TB Centre, Department of Paediatrics and Child Health, Faculty of Medicine and Health Sciences, Stellenbosch University, Cape Town, South Africa
$^2$Centre for Infectious Disease Epidemiology and Research (CIDER), School of Public Health and Family Medicine, Faculty of Health Sciences, University of Cape Town, Observatory, South Africa
$^3$Health Intelligence Directorate, Department of Health and Wellness, Western Cape Department of Health, Cape Town, South Africa
$^4$Division of Infectious Disease and Tropical Medicine, Centre for Infectious Diseases, Heidelberg University Hospital, Heidelberg, Germany
$^5$School of Human Sciences, Faculty of Education, Health and Human Sciences, University of Greenwich, London, UK
$^6$Faculty of Epidemiology and Population Health, London School of Hygiene & Tropical Medicine, London, UK

**Acknowledgements**  We wish to acknowledge our implementing partners; the University of Cape Town and the Centre for Infectious Disease Epidemiology and Research (CIDER) in the Western Cape Province. We further acknowledge the staff at the Western Cape Provincial Health Data Centre (PHDC) for their invaluable assistance. We highly appreciate input from the health staff at the provincial, district and subdistrict health offices and those at the facilities in which the study was implemented.

**Contributors**  S-AM and MO planned and designed the study, were involved in data collection, extraction and validation. MO produced the analysis. S-AM and MO interpreted the data. S-AM wrote the first draft and finalised the submission based on author and peer reviewer input. AvD, RD, FP and MS were involved in data collection, extraction and validation, interpretation of results, reviewed the manuscript and provided critical input. ACH, FMM, JAH, PB and AB-T were involved in data interpretation, reviewed the manuscript and provided critical input. S-AM is the guarantor. All authors have reviewed the final version of the manuscript and approve of its content and submission for publication.

**Funding**  This research study and publication was supported by the Bill and Melinda Gates Foundation (BMGF), Investment INV-007130. The contents are the responsibility of the authors and do not necessarily reflect the views of the BMGF. The funders had no role in study design, data collection and analysis, decision to publish or preparation of the manuscript. ACH is funded by the South African National Research Foundation SaRchi Chair in Paediatric Tuberculosis.

**Competing interests**  None declared.

**Patient and public involvement**  Patients and/or the public were not involved in the design, or conduct, or reporting, or dissemination plans of this research.

**Patient consent for publication**  Not applicable.

**Ethics approval**  Stellenbosch University Health Research Ethics Committee (N18/07/069). The study included routine programmatic data and so did not include human participants directly.

**Provenance and peer review**  Not commissioned; externally peer reviewed.

**Data availability statement**  Data are available upon reasonable request. The de-identified data set is available from the first author, on request. This analysis was nested within a broader study. There are sub-analyses that are ongoing and the data set will only be made public after all the sub-analyses have been completed and published.

**ORCID iD**
Sue-Ann Meehan http://orcid.org/0000-0002-0826-1833

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
