## [Reviewer comments · BMJ Open]

ARTICLE DETAILS

TITLE (PROVISIONAL)	The association between tuberculosis and pregnancy outcomes: A retrospective cohort study of women in Cape Town, South Africa.
AUTHORS	Meehan, S; Hesselning, A C; von Delft, Arne; Marx, Florian M.; Hughes, Jennifer A.; Bock, Peter; Banke-Thomas, Aduragbemi; Dunbar, R.; Phelanyane, Florence; Smith, Mariette; Osman, Muhammad

VERSION 1 – REVIEW

REVIEWER	Marian Loveday South African Medical Research Council
REVIEW RETURNED	28-Oct-2023

GENERAL COMMENTS	Pregnant women with tuberculosis have high rates of unfavourable pregnancy outcomes irrespective of HIV status in Cape Town, South Africa Manuscript ID: bmjopen-2023-081209 Thank you for asking me to review this all-important manuscript, which is very well written and provides evidence on the effect of TB in a poorly documented population. In the manuscript the authors use routine data from 2 sub-districts to report TB treatment and pregnancy outcomes and determine risk factors associated with an unfavourable pregnancy outcome for women with TB. General concerns: This manuscript focusses on treatment and pregnancy outcomes in women who had an episode of TB during their pregnancy. Proportions of unfavourable outcomes are documented, risk factors identified and the conclusion of the abstract is 'Pregnant women with TB are a highly vulnerable group who need additional support prior to, during and after TB treatment to improve TB treatment outcomes and pregnancy outcomes, regardless of HIV status.' However, what is lacking is a comparator. Is there any data from the data set used reporting treatment outcomes in women of reproductive age who were not pregnant? Similarly, is there any data detailing pregnancy outcomes in women who did not have TB? Even if there are only treatment outcomes for all women of reproductive age, irrespective of whether they were pregnant or not, and pregnancy outcomes in all women irrespective of TB, a comparison would be useful and interesting to address in the discussion. Similarly, was ILTU higher in pregnant vs non-pregnant women? The lack of a comparator group is a limitation of this study and should be included as a limitation. Line 331 – 336: This study was nested within the 'LINKEDin' study which aimed to reduce initial loss to follow-up. However, this is not
---

	the focus of this manuscript. Therefore, the detailed explanation (lines 331 – 338) of initial lost to follow-up (ILFU) is unnecessary as ILFU has already been adequately explained. Similarly paragraph 340 – 353 should be shortened to focus on pregnant women and ILFU. Also in the conclusion the only finding emphasised is the high proportion of women diagnosed with TB who never started TB treatment, whereas there were other important findings, such as the late booking into antenatal care. First paragraph of the discussion: Usually in the first paragraph of the discussion, the study question raised in the final paragraph of the introduction is answered. Given that the study question was to determine risk factors associated with unfavourable pregnancy outcomes for women with TB, the first paragraph of the discussion needs to be revised. The details of HIV services is not the focus of this manuscript. The use of language: In several places the authors use the term ‘vulnerable population,’ which, together with ‘at risk’ and ‘special’ population is no longer considered acceptable. Also, where possible, rather than writing ‘people infected with TB or HIV’, ‘people living with TB and HIV’ is now considered more acceptable. Line 355: ‘Most women had their pregnancy recorded only after they were diagnosed with TB.’ Can you discuss this, as this is not the situation in all settings, where TB is often first diagnosed when the woman comes for antenatal care. Lines 392-393: ‘In the Cape Town Metro, 75% of women attend antenatal services before 20 weeks.’ This contrasts with your findings, where you report that a third of the pregnant women only reported to antenatal care 30 days before delivery. Possible reasons for this need to be discussed. Lines 339-441: ‘Pregnant women with TB are a highly vulnerable group who need additional support prior to, during and after TB treatment to reduce poor pregnancy outcomes, even those who are HIV-negative.’ In the lines above and several other places in the manuscript you note that additional support and strategies are needed. Given the work you have done, what support and strategies would you recommend/suggest? A couple of typos Line 68: otucomes should be outcomes Line 130: pregnancy outcome should be outcomes Line 313: internventions should be interventions
--	--

REVIEWER	Sefineh Feleke Woldia University, Public health
REVIEW RETURNED	07-Nov-2023

GENERAL COMMENTS	It is a well-written manuscript. Try to improve typographical errors throughout the document. What things did you add after you did this research? What is your source population? How do you select variables?
--

VERSION 1 – AUTHOR RESPONSE

Reviewer 1:

General concerns:

1. This manuscript focusses on treatment and pregnancy outcomes in women who had an episode of TB during their pregnancy. Proportions of unfavourable outcomes are documented, risk factors identified and the conclusion of the abstract is 'Pregnant women with TB are a highly vulnerable group who need additional support prior to, during and after TB treatment to improve TB treatment outcomes and pregnancy outcomes, regardless of HIV status.' However, what is lacking is a comparator. Is there any data from the data set used reporting treatment outcomes in women of reproductive age who were not pregnant? Similarly, is there any data detailing pregnancy outcomes in women who did not have TB? Even if there are only treatment outcomes for all women of reproductive age, irrespective of whether they were pregnant or not, and pregnancy outcomes in all women irrespective of TB, a comparison would be useful and interesting to address in the discussion. Similarly, was ILTU higher in pregnant vs non-pregnant women? The lack of a comparator group is a limitation of this study and should be included as a limitation.

Thank you for highlighting this. We acknowledge the limitation of not having a comparator and have included this as a study limitation. This cohort analysis was nested within a larger study and our data extract was restricted to persons with TB. We were not able to compare pregnancy outcomes for women who did not have TB during their pregnancy. This has been proposed as future work. In this paper, we have presented the effect of TB outcomes on pregnancy. We further recognise the importance of a comparison by pregnancy status and while we are not able to present this in this study, we have used this as an opportunity to highlight the need for future work to compare outcomes for women by pregnancy status. Lines 463-468.

2. Line 331 – 336: This study was nested within the 'LINKEDin' study which aimed to reduce initial loss to follow-up. However, this is not the focus of this manuscript. Therefore, the detailed explanation (lines 331 – 338) of initial lost to follow-up (ILFU) is unnecessary as ILFU has already been adequately explained. Similarly, paragraph 340 – 353 should be shortened to focus on pregnant women and ILFU. Also, in the conclusion the only finding emphasised is the high proportion of women diagnosed with TB who never started TB treatment, whereas there were other important findings, such as the late booking into antenatal care.

We have deleted the explanation of ILTFU as suggested (now lines 359-369). We have shortened the paragraph as suggested. See lines 352-358 and have reworded the conclusion (lines 477-490).

3. First paragraph of the discussion: Usually in the first paragraph of the discussion, the study question raised in the final paragraph of the introduction is answered. Given that the study question was to determine risk factors associated with unfavourable pregnancy outcomes for women with TB, the first paragraph of the discussion needs to be revised. The details of HIV services is not the focus of this manuscript.

I have revised the first paragraph of the discussion, as per your recommendation. See lines 308-311.

4. The use of language: In several places the authors use the term 'vulnerable population,' which, together with 'at risk' and 'special' population is no longer considered acceptable. Also, where possible, rather than writing 'people infected with TB or HIV', 'people living with TB and HIV' is now considered more acceptable.

Thank you for picking this up. We have addressed this terminology and revised throughout. The conclusion refers to 'pregnant women are a complex population...' (lines 482-483). This is in line with the proposed new terminology.

In addition, we have rethought the terminology used to describe pregnancy outcomes and have opted to rephrase these as 'Good' and 'Adverse' pregnancy outcomes (originally phrased as 'favourable' and 'unfavourable'). We think that the term 'adverse' is a better description for

a composite term that includes mortality. This also aligns well with much of the literature. We have made these changes throughout the manuscript.

5. Line 355: 'Most women had their pregnancy recorded only after they were diagnosed with TB.' Can you discuss this, as this is not the situation in all settings, where TB is often first diagnosed when the woman comes for antenatal care.

We have added an explanation of why we think we may have found this result. Please see lines 391-397.

6. Lines 392-393: 'In the Cape Town Metro, 75% of women attend antenatal services before 20 weeks.' This contrasts with your findings, where you report that a third of the pregnant women only reported to antenatal care 30 days before delivery. Possible reasons for this need to be discussed.

This finding may be related to the limitations posed by the use of routine data that 'the capturing of the first date known to the antenatal services may not always be accurate'. This has been highlighted in lines 394-397.

7. Lines 339-441: 'Pregnant women with TB are a highly vulnerable group who need additional support prior to, during and after TB treatment to reduce poor pregnancy outcomes, even those who are HIV-negative.' In the lines above and several other places in the manuscript you note that additional support and strategies are needed. Given the work you have done, what support and strategies would you recommend/suggest?

We have made some recommendations on what strategies could be used to support this population to link to care, based on our previous work. See lines 373-381.

8. A couple of typos
Line 68: otucomes should be outcomes
Line 130: pregnancy outcome should be outcomes
Line 313: internventions should be interventions

Thank you for picking this up. We have revised accordingly.

Reviewer 2:

1. Try to improve typographical errors throughout the document.

Thank you for highlighting this. We have re-read the paper and made changes to address typographical errors.

2. What things did you add after you did this research?

We have used the findings from this study as preliminary data, for a larger grant application that is looking to improve the care pathway for pregnant and postpartum women with TB and TB/HIV, which would ultimately improve TB treatment and pregnancy outcomes for this population.

3. What is your source population?

Our source population was all females diagnosed with TB at either a hospital or PHC facility in the 2 subdistricts of the Western Cape Province between October 2018 and March 2020, as recorded in the Provincial Health data Centre. (Lines 169-172).

4. How do you select variables?

We were firstly guided by the literature and then by the availability of variables within the routine data extract that we had. The extract included all programmatic indicators collected for TB, HIV and maternal services. We defined favourable/unfavourable TB treatment and

pregnancy outcomes, using relevant variables from the dataset.

VERSION 2 – REVIEW

REVIEWER	Marian Loveday South African Medical Research Council
REVIEW RETURNED	16-Dec-2023

GENERAL COMMENTS	Pregnant women with tuberculosis have high rates of unfavourable pregnancy outcomes irrespective of HIV status in Cape Town, South Africa Manuscript ID: bmjopen-2023-081209 Thank you for addressing the comments I made previously. I still have one concern about your discussion and conclusion. In a couple of places, the authors state that pregnancy outcomes in women with TB are a cause of 'major concern'. I understand from your revisions that your study had no comparator group, but there are other South African studies which report pregnancy outcomes, in women living with and without HIV. In the discussion the authors state 'HIV-positive women were significantly more likely to have a stillbirth or premature infant compared to HIV-negative women(43) and were at increased risk of having a 'small for gestational age' infant, regardless of when they had initiated ART(44)'. However, there are no details of the proportion of stillbirths, infants born prematurely or with low birth weight in either of these studies to support the authors statement that pregnancy outcomes for women with TB are a cause of 'major concern'. These details, or details from other studies are necessary to support the authors conclusion that pregnancy outcomes in women with TB are a cause of 'major concern'.
---

VERSION 2 – AUTHOR RESPONSE

Reviewer 1:

Thank you for addressing the comments I made previously. I still have one concern about your discussion and conclusion. In a couple of places, the authors state that pregnancy outcomes in women with TB are a cause of 'major concern'. I understand from your revisions that your study had no comparator group, but there are other South African studies which report pregnancy outcomes, in women living with and without HIV. In the discussion the authors state 'HIV-positive women were significantly more likely to have a stillbirth or premature infant compared to HIV-negative women (43) and were at increased risk of having a 'small for gestational age' infant, regardless of when they had initiated ART(44)'. However, there are no details of the proportion of stillbirths, infants born prematurely or with low birth weight in either of these studies to support the authors statement that pregnancy outcomes for women with TB are a cause of 'major concern'. These details, or details from other studies are necessary to support the authors conclusion that pregnancy outcomes in women with TB are a cause of 'major concern'.

The references 43 (Cornish etal) and 44 (Malaba etal) specifically focus on pregnancy outcomes among pregnant women living with and without HIV. It is not reported whether these women had TB or not. The systematic review (reference 42; Portwood etal), also does not report if women had TB during their pregnancy, but this is a good reference for pregnancy outcomes by HIV status and ART status. We have revised the paragraph by deleting references 43 and 44 and adding reference 15

(Bekker et al), which supports our conclusion. We have added the detail on prematurity, low birth weight and deaths from the Bekker et al paper to further support our conclusion that 'pregnancy outcomes in women with TB are a cause of 'major concern'. See lines 372-384 (in the tracked version).